# Fracture Healing in 37 Dogs and Cats with Implant Failure after Surgery (2013–2018)

**DOI:** 10.3390/ani13091549

**Published:** 2023-05-05

**Authors:** Timothy L. Menghini, Georgia Shriwise, Peter Muir

**Affiliations:** Comparative Orthopaedic Research Laboratory, Department of Surgical Sciences, School of Veterinary Medicine, University of Wisconsin-Madison, Madison, WI 53706, USA

**Keywords:** implant failure, biomechanics of fracture fixation, internal fixation, plate, area moment of inertia

## Abstract

**Simple Summary:**

Identifying risk factors for implant failure in small animal orthopedics could improve outcomes in clinical patients. Abnormal fracture healing has been hypothesized to correlate with implant failure. This study retrospectively reviewed a consecutive series of implant failure cases in dogs and cats. Area moment of inertia (AMI), plate working length, and bone screw density were determined as appropriate for each case. Implant failure occurred in 23% of the fracture cases in the study period and was associated with an increased risk of delayed union, malunion, or non-union of the fracture. Major complications were found in 56% of the fracture repairs and were associated with delayed union. Surgical revision was performed in 49% of implant failure cases. Common problems were implant loosening and failure of low AMI plates. Implant AMI should routinely be considered during preoperative planning.

**Abstract:**

Implant failure is common in small animal orthopedics, but risk factors are rarely reported. Our objective was to determine whether abnormal fracture healing was associated with implant failure after fracture fixation in dogs and cats in a consecutive series of cases. Thirty-seven client-owned animals (thirty-two dogs, five cats) diagnosed with implant failure after fracture treatment from January 2013–September 2018 were studied. Medical and radiographic records were retrospectively reviewed to identify patients that underwent fracture fixation using open reduction and internal fixation with subsequent radiographic evidence of implant failure. Area moment of inertia (AMI), plate working length, and bone screw density were determined. Implant failure was found in 39 fractures in 37 animals, representing 23% of fracture cases during the study period. Cases of implant failure were at increased risk of delayed union, malunion, or non-union (*p* < 0.0001). The most common cause of implant failure was loosening (54%); the second most common was plate failure that included low AMI locking plates (28%). Major complications found in 22/39 fractures (56%) were associated with delayed union (*p* < 0.01). Surgical revision was performed in 49% of implant failure cases. Complications were most frequently identified after treatment of humeral fractures (26%). We conclude mechanical failure of implants increases the risk for delayed or abnormal fracture healing and often requires revision surgery. Implant AMI should be considered during preoperative planning. Locking plates are associated with implant failure if plate bending stiffness is not sufficient, based on findings from this case series.

## 1. Introduction

Orthopedic implant failure is defined as either failure of the bone–implant interface resulting in loosening and fracture instability, or mechanical failure, such as implant bending or breakage [1]. Implant failure arises either because of mechanical factors associated with construct loading, biological factors associated with bone healing, or a combination of both [2]. Plastic deformation or breakage of implants occurs because cyclical loading causes implant fatigue [3] or, less frequently, due to an acute high-load event. With larger cyclic-bending loads implant fatigue life is shortened.

Implant bending strength is defined by geometry and material properties. Area moment of inertia (AMI), or the second moment of area, is an important implant feature [4] that is a mathematical representation of how an area is distributed across an object’s central axis in cross section. Bending stiffness of an object is described by the product of area moment of inertia and modulus of elasticity. Most orthopedic implants in veterinary medicine are made of 316L stainless steel alloy. Therefore, implant AMI is proportional to bending stiffness for implants made of the same material. Small changes in implant dimensions can have large effects on bending stiffness.

Most fracture constructs fail at weak points in the implants where the AMI is reduced or where stresses are concentrated, such as a plate hole located over a fracture line. Implants applied to comminuted fractures in a bridging fashion must have higher bending stiffness to resist greater axial loading in the absence of load sharing to prevent fracture. Patient overactivity is also an important risk factor for implant failure.

Implant failure incidence is ~5–19% depending on the bone affected, the fracture configuration, and the fixation method [5,6,7,8]. In dogs with comminuted tibial fractures, complications such as valgus malalignment were found [9]. Femoral or tibial fractures in puppies treated with elastic plate osteosynthesis were also associated with implant failure [5]. Treatment of appendicular fractures in dogs and cats with locking compression plates (LCP) was also associated with implant failure [7]. However, construct failure has not been identified in other studies using non-locking plating [8]. In humans, the incidence of implant failure after fracture fixation is estimated to be <5% [10], with common causes of implant failure including re-injury (87.8%) and infection (2.4%) [11].

The purpose of this retrospective study was to review a consecutive series of appendicular fractures treated with open reduction and internal fixation (ORIF) for evidence of implant failure to determine incidence and mode of implant failure. We expected to find abnormal fracture healing was common in cases with implant failure. We were particularly interested in evaluating construct features, such as AMI, plate working length, and bone screw density that could contribute to implant failure, as this type of risk can be mitigated by implant selection during preoperative surgical planning.

## 2. Materials and Methods

### 2.1. Inclusion Criteria

Medical records and radiographs of cats and dogs that underwent appendicular fracture repair between January 2013 and September 2018 at the UW Veterinary Care Hospital, University of Wisconsin-Madison, were reviewed retrospectively. Cases without radiographic follow-up to confirm radiographic union or implant failure were excluded.

### 2.2. Collection of Perioperative Data

For each case, data included: species, sex, breed, age (months), bodyweight (kg), body condition (1–9, 1 = emaciated, 9 = obese), bone fractured, number of fractures, fracture location, and fracture pattern. Physeal fractures were classified according to Salter and Harris [12]. Concurrent injuries were recorded if present. Open fractures were classified using the Gustilo and Anderson classification scheme [13]. Results of bacterial culture and sensitivity testing and use of bone graft were recorded. In cases with multiple fractures, each fracture was considered separately. Time to diagnosis of implant failure was recorded in days. Time from fracture diagnosis to the first surgery was also considered.

### 2.3. Surgical Stabilization Method

All fracture repairs were performed using ORIF and AO/ASIF principles. Implants used for fracture repair and their associated dimensions were recorded. For plate fixation, the number of cortices above and below the fracture were determined radiographically. Plates were classified as compression, neutralization, or bridging, depending on quality of bone column reconstruction with interfragmentary compression and adequacy of load sharing between the plate and bone [14] after review of radiographs and the surgical report. Plate working length (PWL), the distance between the screws proximal and distal most adjacent to the fracture, plate span (ratio between plate length and bone length), plate span ratio (ratio between plate length and overall fracture length), bone screw density (BSD) (proportion of plate holes with screws), whether the fracture was reconstructable, and number of bone cortices per plate engaged with screws above and below the fracture were also determined.

### 2.4. Assessment of Complications

Postoperative complications were classified as major if surgical treatment for implant failure, including removal of loose or broken implants after clinical fracture healing, or treatment with external coaptation or amputation was performed. Cases with loose or broken implants identified after clinical healing of the fracture that were not revised surgically or treated with coaptation were classified as minor complications. Based on review of the clinical and radiographic findings, cases were subjectively classified as a mechanical failure or a biological and mechanical failure. This classification was based on radiographic evidence of delayed or nonunion fracture, evidence of orthopedic infection confirmed by culture, or a history of chronic corticosteroid medication.

### 2.5. Radiographic Outcome Measures

Bone union was defined as radiographic evidence of bridging callus at ≥3 cortices. Implant failure was diagnosed if broken, bent, or loose implants were identified. Cases were also assessed for development of malunion (malalignment of bone ends in a healed fracture) [15], delayed union (bone healing beyond 12 weeks in adult animals and beyond 6 weeks in immature animals) [16], or nonunion (a fracture that failed to progress to osteosynthesis regardless of healing time [17].

### 2.6. Implant Area Moment of Inertia

Area moment of inertia was calculated for the failed implants using a standard approach [4]. Plate AMI was calculated using the formula: *I* = *bh^3^/12*, where *b* = plate dimension parallel to axis around which moment area of inertia is being calculated and *h* = plate dimension parallel to the applied bending load. The calculations assume a perfect rectangular cross section for plates (See Appendix A).

### 2.7. Statistical Analysis

The D’Agostino–Pearson normality test was performed on all data. Normally distributed data were summarized as mean ± standard deviation. Non-parametric data were summarized as median and range (lowest, highest). Two-tailed Fisher’s exact test was used to test for association between construct failure cases with a major complication and the type of fracture healing that developed. Results were considered significant at *p* < 0.05.

## 3. Results

### 3.1. Study Population

In the study period, 217 patients consisting of 198 dogs and 19 cats were presented for treatment of a fracture (Figure 1). Fifty-nine cases (27%) did not return for radiographic follow-up. In the remaining 158 cases, radiographic union (117 cases, 74%), delayed union (18, 11%), malunion (14, 9%), and nonunion (9, 6%) was found. Construct failure was associated with abnormal fracture healing (*p* < 0.0001, Table 1, odds = 16.7).

### 3.2. Construct Failure Population

We found evidence of construct failure in 32 dogs and 5 cats (23.4%) with 39 failed fracture repairs. The median number of fractures per case was 1 (1, 3), with 32 of 37 cases (86.5%) having a single fracture. There were 10 humeral (25.6%), 8 antebrachial (20.5%), 8 tibial (20.5%), 8 femoral (20.5%), 2 pelvic (5.1%), 2 fibular (5.1%), and 1 metatarsal fracture (2.6%) (Table 2).

Dog breeds were mixed breed (9), Labrador retriever (7), Rhodesian ridgeback (2), Pomeranian (2), Chihuahua (2), Yorkshire terrier (2), and 1 each of fox terrier, French bulldog, golden retriever, Mi-Ki, rat terrier, Shetland sheepdog, Staffordshire terrier, and whippet. Cat breeds were domestic short hair (3), domestic medium hair (1) and Abyssinian (1). The sex distribution was 12 castrated males (32%), 11 males (30%), 10 females (27%) and 4 ovariohysterectomized females (11%). Median weight for dogs was 11 kg (1.12, 54) and 4.05 kg (0.91, 4.93) for cats. Median body condition score was 5 (3.5, 9). Median age was 12 months (2, 144).

Median time from fracture diagnosis to first surgery was 2 days (0, 25) (n = 36). In one dog with a chronic fracture, this time could not be determined. There were three other fractures where the time from fracture diagnosis to surgery was >7 days. Median time to diagnosis of implant failure was 45 days (10, 1315). Construct failure occurred in 27 reconstructable simple fractures (69%) and 12 comminuted fractures (31%), 7 of which were reconstructable. All comminuted fractures were treated with bone plate fixation. Simple fractures were long oblique (12), transverse (7), short oblique (5), avulsion (2), and spiral (1). Failures included 15 physeal fractures (38%) including Salter–Harris type I (2), II (5) and type IV (8) fractures. Open fractures (10%) were Grade 1 (3) and Grade 2 (1). There were 12 articular fractures (31%).

Mode of failure is summarized in Table 2. Treatment included plate/screw fixation (23) with plates applied to the typical bone surface (Table 3). In the remaining 16 fractures, implants were Kirschner wires (K wire) or Steinmann pins alone (6) or combined with another fixation. Construct failure was considered mechanical in 31 fractures and mechanical/biological in 8 fractures. Major complications were found in 22 fractures (56%) in 21 cases, 19 of which were surgically revised. Delayed union healing was associated with development of a major complication (*p* < 0.01, Table 4). Minor complications were found in 17 fractures (44%) in 17 cases, none of which underwent surgical revision. Malunion healing was associated with development of a minor complication (*p* < 0.001, Table 4). Implant failure incidence rate was 0.234 (37/158 cases with 39 fractures). Incidence rate of major complications in the implant failure group was 0.56, 22/39 fractures in 21 cases. Implant failures included plate bending/breakage (11), screw bending/breakage (5), pin/wire bending/breakage (2), or pin/screw loosening (21). Bacterial culture and sensitivity testing were performed in 7/39 fractures (17.9%); 5 were culture positive. Bone grafting was used in 8 fractures (21%) including cancellous autograft (6), osteoallograft (1), and demineralized bone matrix (1).

### 3.3. Plate Failure (11 Fractures)

Summary data on plate constructs are presented in Table 3. Plate length was 9 ± 2.4 holes, AMI was 0.79 mm^4^ (0.03, 46.7), BSD was 0.83 (0.55, 1), PWL was 10.8 mm (3.9, 72.9), plate span was 55.6 ± 19.6%, and plate span ratio was 4.8 (2.2, 46.0). The number of cortices above the fracture in the proximal fragment was 7 (4, 10) and the number of cortices below the fracture in the distal fragment was 6 (4, 12).

Breakage of mini plates at the level of the fracture occurred after treatment of diaphyseal antebrachial fracture in three dogs (Figure 2). Plate breakage was found after treatment of a Salter–Harris type IV lateral condylar fracture with a 2.4 mm lag screw and a 1.3 mm titanium plate in a dog (Figure 3). Plate failures also included two comminuted tibial diaphyseal fractures in dogs treated with bridging plate fixation including bending of a 3.5 mm dynamic compression plate (DCP) (Figure 4). Plate breakage was also found after treatment of a comminuted femur fracture with a 1.5/2.0 mm bridging LCP in a dog (Figure 5. Median AMI of the 11 plates that failed was 0.61 mm^4^ (0.16, 32.94).

### 3.4. Screw Breakage (Five Fractures)

Breakage of 2.0 mm or 2.7 mm screws was found after treatment of three articular fractures. Breakage of 2.0 mm screws was also found after treatment of multiple proximal metatarsal fractures with partial tarsal arthrodesis and with treatment of a diaphyseal femur fracture in a puppy with a 2.0 mm limited-contact dynamic compression plate (LC-DCP).

### 3.5. Pin/Wire Breakage or Bending (Two Fractures)

Tension band wire breakage and proximal displacement of the ulna apophysis was found after treatment of a Salter–Harris type II fracture in a dog. Pin bending was found after cross pin treatment of a distal femoral Salter–Harris type II fracture in a cat with other ipsilateral major injuries.

### 3.6. Implant Loosening (21 Fractures)

Loosening of the transcondylar lag screw was found in seven lateral humeral condylar fractures in six dogs. Epicondylar crest fixation consisted of a single Kirschner wire (5), two Kirschner wires (1) and a lateral neutralization plate (1). Screw loosening was also observed after plate treatment of a fracture in three cats and two dogs with associated Kirschner wire loosening in one case. Intramedullary pin migration was found in one dog with pin-plate construct treatment of a comminuted fracture. Pin bending or migration was also found after treatment of five physeal fractures and a distal metaphyseal fracture of the fibula. Screw pullout was identified after fracture repair in two dogs.

Additional clinical findings and the specific constructs associated with implant failure are also reported in a Appendix A.

## 4. Discussion

Construct instability is a common complication after fracture repair in small animals [18]. Implant failure can lead to lameness, abnormal healing, prolonged hospitalization, and increased morbidity associated with surgical revision or implant removal. The incidence rate of implant failure in the present study was 0.234; higher than previously reported [4,5,6,7,8]. Risk of abnormal fracture healing was increased in cases with implant failure with an incidence rate of major complications after implant failure of 0.56. We also found that risk of delayed union was increased in patients with major complications from construct failure. Malunion healing was also common in patients with a minor complication associated with implant failure. These observations highlight the importance of preoperative planning.

Implant failure is associated with both mechanical and biological risk factors. We found mechanical failure was common (79%), with both mechanical/biological factors explaining the remaining cases. In human, nonunions account for 4.9% of fractures [19], similar to our study (5.7%), although delayed union was the most common type of abnormal healing we identified after implant failure. Bone grafting was used for initial treatment in only 8 of 39 fractures (21%) and could have been incorporated into preoperative planning more frequently. Autogenous cancellous bone graft can accelerate bone healing by several weeks [20].

AMI is an important consideration in preoperative planning [4]. Small changes in implant dimensions can have large effects on bending stiffness. Plate failure alone or in combination with other implants was the primary mode of failure in 10 bridging and neutralization constructs, particularly when used to treat a comminuted fracture. Our results suggest that it is common to select implants with an AMI that is too small during preoperative planning and consideration of other relevant factors such as biomechanical forces acting on the fracture construct, anatomical location, and fracture configuration.

Many locking implants are used for small animal fracture fixation [21,22] and may be advantageous if the expected failure mode is screw pullout. Lack of familiarity with locking plate mechanical properties may lead to errors in preoperative planning. Locking implants are fixed-angle constructs that are not necessarily stronger in bending than similar sized non-locking plates. Our results suggest this is a particular problem with mini-implants (Appendix A). Plate breakage was identified in several toy breed dogs after antebrachial or femoral fracture repair with plate application to the typical bone surface. Use of a 1.5 mm straight plate with round holes and a small AMI (0.16 mm^4^) was often associated with construct failure. Other 1.5 mm implants, such as the 1.5 mm LCP (AMI = 0.35 mm^4^) [23] or the 1.5/2.0 mm LC-DCP (0.79 mm^4^) have higher bending stiffness and the LC-DCP could also enable interfragmentary compression. Breakage of the 1.5/2.0 mm LCP was also common. Use of a plate with a relatively small AMI may have influenced this outcome. Other mini non-locking plates, such as the 2.0/2.7 mm veterinary cut-to-length plate or the 2.0/2.4 mm LC-DCP, have much larger AMIs (2.0 and 2.66 mm^4^, respectively). Lack of interfragmentary compression in reconstructable fractures may also contribute to high plate strains if a narrow gap remains at the fracture site.

Screw diameter alone should not dictate plate selection. Rather, all elements of a patient fracture assessment which considers clinical, biological, and mechanical factors should be considered [24], together with bone size and whether the plate will be applied to the tension surface of the bone. PWL may influence construct behavior under cyclic loading. However, a recent ex vivo study of PWL using a canine femoral gap model found no significant effects on stiffness, gap motion, and resistance to fatigue [25]. Construct stiffness can also be increased by use of a plate–rod construct [26] or by stacking plates such as veterinary cut-to-length plates. We did not identify any relationships between plate construct failure and bone screw density, plate span, and plate working length in this study.

Implant loosening affected 54% of construct failures. However, few fractures in this series had clinical and radiographic evidence of implant-associated infection although two open fracture cases were culture-positive for methicillin-resistant *Staphylococcus aureus* as described in other reports [27,28,29]. This suggests most implant loosening had a mechanical etiology. Not all construct failure fracture cases underwent initial bacterial culture, so the proportion of culture-positive fractures with construct failure could have been higher.

In this case series, young animals, small patients, and physeal fractures were overrepresented. Physeal fractures can be difficult to repair, particularly humeral condylar physeal fractures. Loosening of the transcondylar lag screw was a common event in this case series. Use of a transcondylar screw with a large core diameter is recommended to mitigate this risk [4]. Use of an anti-rotational K wire, instead of a plate, results in a higher risk of complications [30], an observation that is supported by our results. The dog with a broken lateral epicondylar crest plate (1.3 mm titanium orbital rim plate) was a small Chihuahua. Titanium is inherently less stiff compared to steel; a 1.5 mm steel implant is available as an alternative.

This study had several limitations. Some follow-up data were missing and 27% of cases in the study period did not return for radiographic follow up. Results may also have been influenced by the different surgical teams that provided clinical treatment, although all teams were led by a boarded surgeon experienced in small-animal orthopedic surgery. A strength of our study design was that construct failures were reviewed in a consecutive series of fracture cases which yielded an accurate estimate of incidence. Although the principles of preoperative planning are universal, this approach also necessitated review of a variety of fractures clinically. Analysis of a larger number of cases in future work could enable more detailed assessment of the pattern of construct failure with specific fracture types and enable comparison of management approaches, such as selection of implants considering AMI, use of bone grafting, and bacterial culturing for specific fracture types with and without construct failure. Use of ORIF in all cases rather than minimally invasive plate osteosynthesis [31,32] may also have influenced outcomes. Even if appropriately sized, implant failure may occur with excessive patient activity. Given the retrospective nature of this study and lack of clear medical-record information, the potential role of patient activity after surgery in development of construct failure could not be determined. There were four chronic fractures at the time of first surgical treatment. Fractures where surgical treatment was delayed likely have an inherently higher risk of implant-related complications.

## 5. Conclusions

In conclusion, the incidence rate of implant failure in a client-owned population of 217 dogs and cats that underwent fracture treatment was 0.234. Odds of abnormal fracture healing were 16.7 with implant failure. Major complications principally had a mechanical etiology associated with implant loosening or breakage. Implant AMI is an important part of preoperative planning irrespective of whether a locking plate is used. Preoperative planning and client education regarding postoperative care should aim to create a stable fracture construct during the healing period.

## Figures and Tables

**Figure 1 animals-13-01549-f001:**
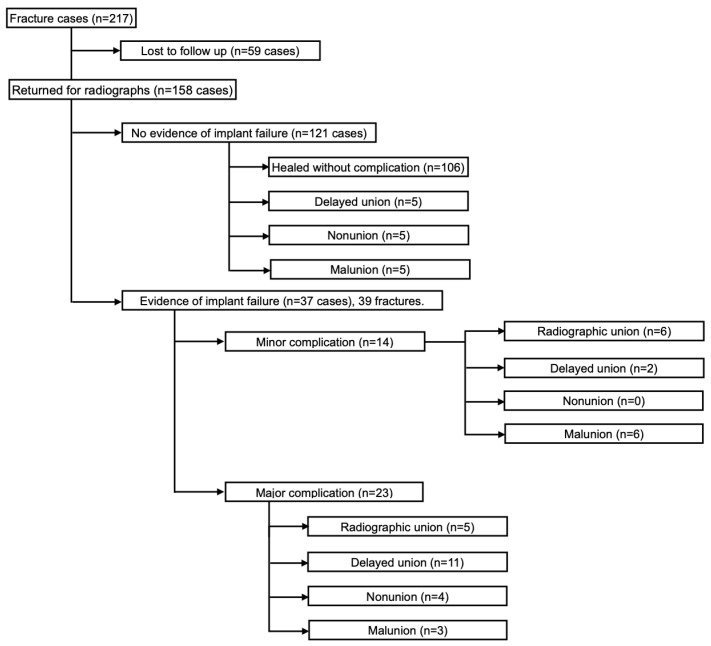
Study flow chart.

**Figure 2 animals-13-01549-f002:**
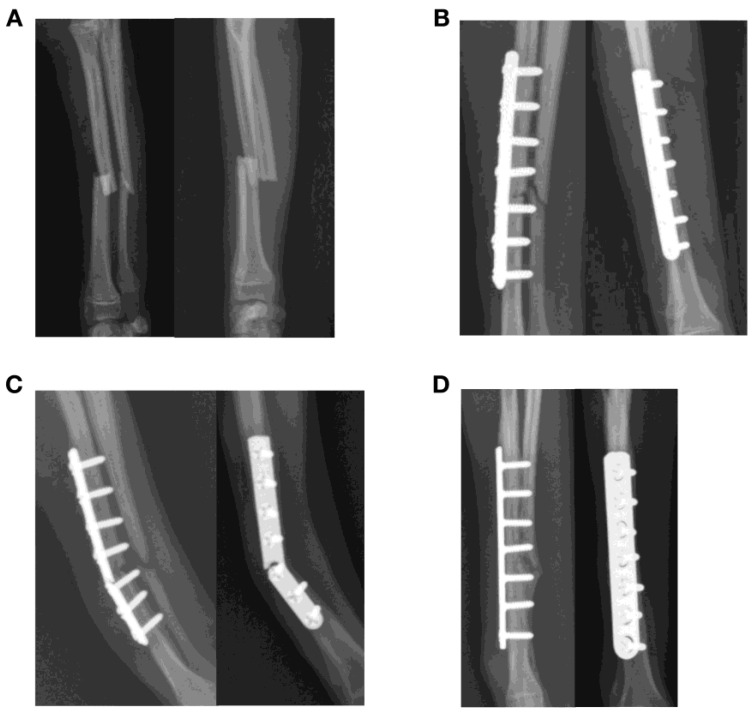
Failure of a plate construct after treatment of a closed transverse fracture of the radius and ulna in a one-year-old Chihuahua. (**A**,**B**) The fracture was stabilized with a 7-hole 1.5 straight plate that did not permit interfragmentary compression, with four 1.5 mm cortical screws proximal and three 1.5 mm cortical screws distal to the fracture site. Autogenous cancellous bone graft was also applied to the fracture. (**C**) Plate breakage was found at 3 weeks after surgery at the level of the fracture. Although the implant was considered a neutralization plate, plate strain at the level of the fracture would be expected to be high after surgery as fracture reduction and the lack of interfragmentary compression left a small gap at the fracture site. (**D**) The fracture construct was revised with another 7-hole 1.5 mm straight plate. The final outcome was a delayed union after 70 days.

**Figure 3 animals-13-01549-f003:**
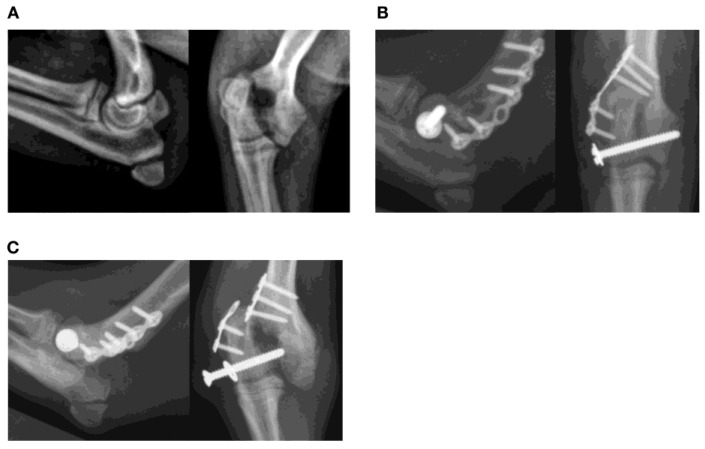
Failure of a neutralization plate construct after treatment of a Salter-Harris IV fracture of the lateral region of the humeral condyle in a 4-month-old Chihuahua mix. (**A**,**B**). The fracture was stabilized with a 2.4 mm transcondylar lag screw and a lateral 6-hole 1.3 mm titanium orbital rim plate. (**C**) Plate breakage at an empty screw hole was found at 4 weeks after surgery despite cage restriction. Revision surgery was declined, and a lateral splint was placed. The final outcome was a delayed union after 76 days.

**Figure 4 animals-13-01549-f004:**
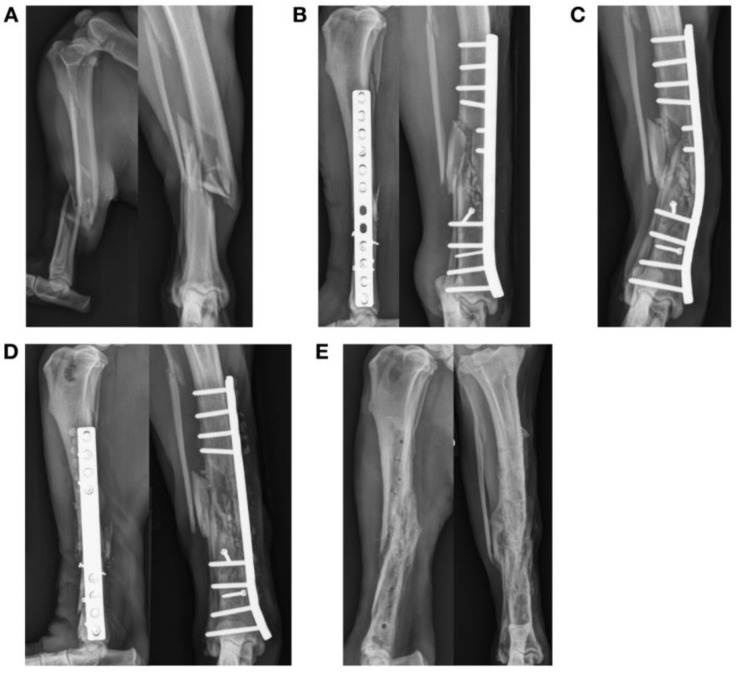
Failure of a bridging plate construct used to treat an open comminuted right tibial diaphyseal fracture in an eight-year-old Labrador retriever (**A**,**B**). The fracture was stabilized with a 12-hole 3.5 mm DCP (AMI = 46.7 mm^4^) and 2 independent 2.4 mm lag screws with 10 cortices engaged above the fracture and 12 cortices below the fracture. Two independent 2.4 mm cortical screws were placed in lag fashion. Two holes over the fracture site were left empty and the fracture was grafted with autogenous cancellous bone graft from the right proximal tibia. Plate span was 72%. The dog was discharged without external coaptation. (**C**) Bending of the plate was found at the level of the fracture and empty screw holes at 2 weeks, causing tibial valgus. (**D**) The fracture construct was revised using an 8-hole 4.5 mm narrow limb-lengthening plate (AMI = 54.9 mm^4^) and the fracture was re-grafted with autogenous cancellous bone. (**E**) Implant removal was performed after clinical union.

**Figure 5 animals-13-01549-f005:**
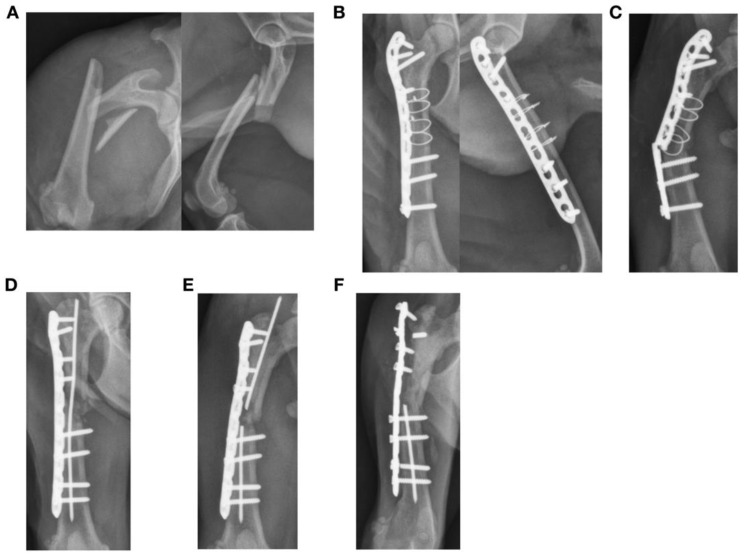
Failure of a bridging plate construct used to treat a closed comminuted left femur fracture in a seven-year-old male intact shih tzu cross. (**A**,**B**) The fracture was stabilized with a 10-hole 1.5/2.0 mm LCP with seven 2.0 mm screws and four loops of 24-gauge cerclage without use of bone grafting. Five cortices engaged the proximal bone segment, and six cortices engaged the distal segment. Three plate holes over the area of comminution were left unfilled. Plate span was 70%. (**C**) Plate breakage at the level of the fracture was identified at 14 weeks. (**D**) The fracture construct was revised with an 0.045″ intramedullary pin and a replacement plate with two empty holes over the fracture. Cancellous bone autograft and omentum were placed in the fracture site. (**E**) Failure of the revised construct was found at 6 weeks. The broken pin was removed, bone morphogenic protein was placed in the fracture site and a Spica splint was applied. (**F**) Further failure of the construct with breakage and loosening of the proximal screws was found at 30 weeks with quadriceps contracture and palpable instability at the fracture site. Amputation was performed at 323 days after treatment started because of nonunion.

**Table 1 animals-13-01549-t001:** Outcome contingency table of 158 cases that returned for first radiographic follow-up after surgery for fracture repair.

	Construct Failure	No Construct Failure	*p* Value
Healed	11	106	0.0001
Abnormal healing	26	15	
Delayed union	13	5	
Nonunion	4	5	
Malunion	9	5	
Total	37	121	

**Table 2 animals-13-01549-t002:** Bone injured and mode of failure in 39 fracture constructs in 37 dogs and cats.

	Mode of Failure				
Bone	Plate Failure	Implant Loosening	Screw Failure	Pin/wire Failure	Total
Radius/ulna	3	3	1	1	8
Humerus	1	7	2		10
Metatarsus			1		1
Tibia	3	5			8
Fibula		2			2
Femur	2	4	1	1	8
Pelvis	2				2
Total	11	21	5	2	39

**Table 3 animals-13-01549-t003:** Biomechanical properties of antebrachial, humeral, tibial femoral, and tarsal fractures treated with a bone plate (n = 23).

Case Number	Bent or Broken Plate	Bone	Location	Comminuted	Reconstructable	Plate Length (Holes)	Number of Cortices in the Proximal Segment	Number of Cortices in the Distal Segment	Bone Screw Density	Plate Span (%)	Plate Span Ratio	Plate Working Length (mm)	Plate Type	AMI (mm^4^)	Surface
30	Y	Radius	Mid-diaphyseal	N	Y	8	10	6	1.00	60	5.2	4.8	1.5 mm straight plate	0.16	Cranial
31	Y	Radius	Proximal diaphyseal	N	Y	8	8	8	1.00	45	9.9	6.0	2.0 mm DCP	0.42	Cranial
32	Y	Radius	Distal diaphyseal	N	Y	7	8	6	1.00	50	21.9	3.9	1.5 mm straight plate	0.16	Cranial
33	N	Ulna	Proximal diaphyseal	Y	Y	7	4	4	0.57	31	5.5	18.5	1.5/2.0 mm cut-to-length	0.58	Lateral
36	N	Ulna	Proximal diaphyseal	Y	Y	7	6	8	1.00	33	6.8	6.6	2.0 mm DCP	1.4	Lateral
37	N	Ulna	Proximal diaphyseal	Y	Y	11	7	7	0.82	37	2.8	15.4	2.4 mm LCP	2.66	Caudal
21	N	Humerus	Diaphyseal	N	Y	8	8	7	1.00	45	28.7	7.4	2.0 mm DCP	1.4	Medial
24	N	Humerus	Bicondylar	Y	Y	9	8	9	0.89	51	4.3	12.2	Double 2.0 mm LCP	2.66	Medial and lateral
26	Y	Humerus	Condylar	N	Y	6	6	4	0.83	36	9.1	5.9	1.3 mm titanium orbital rim plate	0.03	Lateral
28	N	Humerus	Condylar	N	Y	5	4	4	0.80	42	4.2	10.8	2.0 mm DCP	0.42	Lateral
10	N	Metatarsal II, III	Diaphyseal	N	Y	8	6	6	0.75	57	n/a	18.3	2.0 mm DCP *	0.42	Medial and lateral
1	Y	Tibia	Diaphyseal	Y	N	12	10	12	0.83	71.9	3.5	38.5	3.5 mm DCP	32.94	Medial
2	Y	Tibia	Diaphyseal	Y	Y	13	7	8	0.62	95.8	2.6	41.4	1.5/2.0 LCP	0.79	Medial
3	N	Tibia	Diaphyseal	Y	N	11	9	10	0.91	77.1	2.8	26.0	3.5 mm broad DCP	46.7	Medial
11	Y	Tibia	Diaphyseal	Y	N	14	10	9	0.79	62.1	3.2	6.3	Stacked 1.5 mm cuttable plate with spaces (VI)	2.8	Medial
9	N	Fibula	Malleolar	N	Y	5	4	4	0.80	23.1	4.8	9.4	1.5 mm straight plate	0.16	Lateral
14	Y	Femur	Diaphyseal	Y	Y	10	5	6	0.70	70.2	2.8	21.7	1.5/2.0 LCP	0.79	Lateral
15	N	Femur	Diaphyseal	Y	N	9	5	6	0.67	73.2	3.0	17.7	1.5 mm straight plate stacked	0.32	Lateral
16	N	Femur	Diaphyseal	N	Y	10	10	8	0.80	59.8	2.2	30.7	2.0 mm LC-DCP	2.66	Lateral
17	Y	Femur	Diaphyseal	N	Y	9	10	8	1.00	84.5	8.8	10.0	2.0 mm LCP	0.79	Lateral
18	N	Femur	Diaphyseal	Y	N	11	6	6	0.55	83.2	4.7	72.9	3.5 mm LCP	32.94	Lateral
19	Y	Ilium	Shaft	N	Y	7	8 (cranial)	6 (caudal)	1.00	59.9	25.9	8.1	2.0 mm DCP	0.42	Lateral
20	Y	Acetabulum	Caudal	Y	Y	6	6 (cranial)	6 (caudal)	1.00	32.1	46.0	6.7	2.7 mm SOP	n/a	Lateral

Note: Bone Screw density is the number of plate holes occupied by screws divided by total number of holes in the plate. Plate span is the ratio of plate length to bone length. Plate span ratio is the plate length divided by the fracture length. Plate working length is the distance in millimeters between the proximal and distal screws adjacent to the fracture. AMI, Area moment of inertia *I = bh^3^/12* (*b* = plate dimension parallel to axis around which moment area of inertia is being calculated, *h* = plate dimension parallel to the applied bending load). n/a, not applicable. All plates were manufactured by De Puy Synthes, Raynham, MA, USA, except for the SOP plate, which was manufactured by Orthomed, Huddersfield, UK, and the 1.5/2.0 mm cuttable plate with spaces which was manufactured by Veterinary Instrumentation, Sheffield, UK. * Partial arthrodesis was performed using a medial 2.0 mm DCP as well as a lateral 2.0 mm LC-DCP for partial arthrodesis.

**Table 4 animals-13-01549-t004:** Healing response and complications in 37 cases with 39 fracture construct failures.

	Major Complication (n = 22)	Minor Complication (n = 17)	*p*-Value
Radiographic union	6	5	1.00
Malunion	1	9	<0.001
Delayed union	12	2	<0.01
Nonunion	3	1	0.62

## Data Availability

Data are contained within the article or Appendix A.

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
