# Peer review of "Fracture Healing in 37 Dogs and Cats with Implant Failure after Surgery (2013–2018)"

_animals, 2023, doi:10.3390/ani13091549_

Round 1
Reviewer 1 Report
It is an interesting paper, well written, and planned, which deserves to publish in order to proportionate important information in the area of complications of fractures
It is the first study that analyzes implant failure in a consecutive clinical case series. So, it is original and present motives of the implant failure helping to planning correctly the surgery. There are studies of specific fractures o implants but no a consecutive clinical cases with the focus on implant failure.Author Response
Thank you for the positive comment about our article.
Reviewer 2 Report
This is a well-designed, well written retrospective study on fracture construct failure. The study analyses and summarizes clinical and biomechanical parameters which lead and may lead to implant failure. Results and recommendations will be helpful reducing the incidence of such failures in small animal patients. The paper adds to filling an important gap in selecting adequate materials and their proper application in operative osteosynthesis.

Author Response
Thank you for the positive comment. We agree that results of this research will help guide pre-surgical planning for small animal fracture repair.
Reviewer 3 Report
Excellent paper based on a large overall experience. Just few and minor suggetions are needed:
Line 14: Major complications were found
Paragraph 2.2: Along the paragraph probably could be better to indicate if had been all the fractures included operated in a short time after the lesion.
Line 144-145: “Among the remaining 158 cases, radiographic union (117 cases, 74%), delayed union 144 (18, 11%), malunion (14, 9%), and nonunion (9, 6%) were reported” (or similar).
Line 150: Consider substituting the word “initial” with “first” or similar.
Line 166: could have senso to consider median body weight separated for dogs and cats? They are two different species with a consistently different body weight range and consequently little changes in weight has different meaning as mechanical effect. For example consider the difference between cats of 4 and 8 kg and the same throughout the canine breeds.
Table 3: First letters of columns 1 (Case) and 2 (Bent) in line 1 are not fully visible.
Figure 4: Description of image E is missing.
Line 336: Although not all construct failure fracture cases that underwent initial.
Line 353: Consider to substitute “incidence” instead of “incident rate”.
General consideration: Are data available considering quality of care in the post operative period? Is it possible to state if there could be correlation between implant failure and inadequate post-operative care, or how many of the implant failures may be associated to excessive activity after surgery?
Author Response
Thank you for the positive comment about this paper. Responses to your specific comments are provided below.
1). ‘Line 14: Major complications were found’
The writing error has been corrected
2). Paragraph 2.2: ‘Along the paragraph probably could be better to indicate if had been all the fractures included operated in a short time after the lesion’
The meaning of this comment is not fully clear. It seems this comment is asking about the time from fracture diagnosis to initial surgical treatment and whether any patient may have undergone treatment for a chronic fracture that could have great risk of construct complications. To address the manuscript now considers this result. Four of the fractures were chronic at the time of initial treatment (line 94, line 171-173, lines 377-379).
3). Line 144-145: ‘Among the remaining 158 cases, radiographic union (117 cases, 74%), delayed union 144 (18, 11%), malunion (14, 9%), and nonunion (9, 6%) were reported (or similar)’
Thank you for this comment. The writing error has been corrected (line 147).
4). Line 150: ‘Consider substituting the word “initial” with “first” or similar’
Thank you for this comment. The recommended change has been made (line 152).
5). Line 166: ‘could have senso to consider median body weight separated for dogs and cats? They are two different species with a consistently different body weight range and consequently little changes in weight has different meaning as mechanical effect. For example, consider the difference between cats of 4 and 8 kg and the same throughout the canine breeds’
Thank you for the comments. This section of the results has been revised as suggested (lines 167-170).
6). Table 3: ‘First letters of columns 1 (Case) and 2 (Bent) in line 1 are not fully visible’
Thank you for this comment. The formatting of the first row has been corrected.
7). Figure 4: ‘Description of image E is missing’
Thank you for this comment. The problem with the figure legend has been corrected.
8). Line 336: ‘Although not all construct failure fracture cases that underwent initial’
Thank you for this comment. The relevant text has been revised to improve clarity (line 349).
9). Line 353: ‘Consider to substitute “incidence” instead of “incident rate’
Thank you for this comment. The sentence has been revised as recommended (line 366).
10). General consideration: Are data available considering quality of care in the post operative period? Is it possible to state if there could be correlation between implant failure and inadequate post-operative care, or how many of the implant failures may be associated to excessive activity after surgery?
This is a good point. The role of patient activity in construct overload is briefly mentioned in the discussion (lines 374-376). This section has been revised to improve clarity considering this comment. This is likely an important factor but could not be clearly investigated in a records based retrospective study like this.
Reviewer 4 Report
Congratulations on the interesting work
the main question addressed by the research is the Risk factors of Implant failure in small animal orthopaedics;
The reports of treatment secondary problems are rare are rare. It is a retrospective observational study that has a specific structure and a little different from the conventional one, mainly in terms of materials and methods, but it seems to me the most adequate for the purpose and the analysis of the reasons for the problems is also not very common but it is essential to avoid them in the future and improve medical services to animals. Therefore, in general, I think the content is very useful, it will be of interest to professionals in the area and deserves publication.
Compared with other published material, complement the success of treatments that are more frequently reported.
The preoperative planning of fractures implants should improve
The conclusions consistent with the evidence and arguments presented and they address the main question posed. Adequate reoperative planning and client education regarding postoperative care during the healing period are essential for fracture treatment success.
Author Response
Thank you for the helpful comment. We agree that the paper reports interesting data regarding risk factors for implant failure.